# Rapid methods for identifying barriers and solutions to improve access to community health services: a scoping review protocol

Luke Nelson Allen ![ORCID] ,[1] Hagar Azab,[1] Ronald Jonga,[1] Iris Gordon,[1] Sarah Karanja,[2] Jennifer Evans,[1] Nam Thaker,[1] Jacqueline Ramke ![ORCID] ,[1] Andrew Bastawrous[1]

[1]Department for Clinical Research, London School of Hygiene & Tropical Medicine, London, UK
[2]KEMRI, Nairobi, Kenya

**Correspondence to**
Dr Luke Nelson Allen;
drlukeallen@gmail.com

## ABSTRACT

**Objectives** Low attendance rates for community health services reflect important barriers that prevent people from receiving the care they need. Services and health systems that seek to advance Universal Health Coverage need to understand and act on these factors. Formal qualitative research is the best way to elicit barriers and identify potential solutions, however traditional approaches take months to complete and can be very expensive. We aim to map the methods that have been used to rapidly elicit barriers to accessing community health services and identify potential solutions.

**Methods and analysis** We will search MEDLINE, Embase, the Cochrane Library and Global Health for empirical studies that use rapid methods (<14 days) to elicit barriers and potential solutions from intended service beneficiaries. We will exclude hospital-based and 100% remotely delivered services. We will include studies conducted in any country from 1978 to present. We will not limit by language. Two reviewers will independently perform screening and data extraction, with disagreements resolved by a third reviewer. We will tabulate the different approaches used and present data on time, skills and financial requirements for each approach, as well as the governance framework and any strengths and weaknesses presented by the study authors. We will follow Joanna Briggs Institute (JBI) scoping review guidance and report the review using the Preferred Reporting Items for Systematic Reviews and Meta-Analyses Extension for Scoping Reviews.

**Ethics and dissemination** Ethical approval is not required. We will share our findings in the peer-reviewed literature, at conferences, and with WHO policymakers working in this space.

**Registration** Open Science Framework (https://osf.io/a6r2m).

## STRENGTHS AND LIMITATIONS OF THIS STUDY

⇒ As far as we are aware, this will be the first review to evaluate the rapid approaches used to elicit barriers to access community health services and identify potential solutions.
⇒ Improving access and grounding service improvements in community engagement are two major global health priorities.
⇒ Our review will follow best-practice guidelines, use a search strategy devised by an information specialist, and use independent dual review at every stage.
⇒ We will miss rapid approaches that have been used effectively but not written about, and those that take longer than 14 days to deliver findings.

## INTRODUCTION
### Rationale

Many health programmes experience large mismatches between people identified with a clinical need and those who attend services. A recent international systematic review of non-attendance across all medical specialities estimated that 23% of clinic appointments are missed, with the highest rate observed in Africa (43%).[1] Low attendance rates often reflect significant barriers faced by users.[2] Marginalised populations are often the least likely to receive care.[3 4] Improving access to ensure that all individuals and communities receive the care they need lies at the heart of Universal Health Coverage—a core element in the Sustainable Development Agenda.[5 6]

Complex supply and demand factors govern access to health services and multiple frameworks have been developed, typically defining access as the ability to perceive, seek, reach, pay for and engage with care.[2 7–11] Access is increasingly being extended through the use of digital services and remote consultations.[12 13] While these services are useful, they come with their own set of barriers and equity issues, and cannot fully replace the central role played by in-person clinical providers.[12 14] When it comes to identifying barriers to attending in-person clinical services and potential solutions, WHO has noted that 'it is the experts who identify the problems and formulate interventions, while the problems and solutions as perceived by those at particular risk rarely constitute

the base for action'.[15] Efforts to improve attendance rates should be grounded in an understanding of both supply-side and demand-side barriers, elicited through engagement with affected communities.[2 16 17] The WHO Primary Health Care (PHC) Operational Framework defines engagement as 'the process of involving people and communities in the design, planning and delivery of health services, thereby enabling them to make choices about care and treatment options or to participate in strategic decision-making on how health resources should be spent'. Turk and colleagues note that health service interventions 'must be done with, and not simply done to, the people affected'.[18]

Research evidence aligns with common sense in finding that involving communities in the development of services improves health outcomes and sustainability.[18] For-profit enterprises seem to understand the value of engaging with their customers: many companies use focus groups and market research to continually hone their products and services to meet the evolving needs of their customer base.[19] Our sense is that health programmes are less active in this space. Ideally—given the scale of the problem—health system managers would be able to deploy affordable, rapid and methodologically sound tools to engage with the groups that face the highest barriers to accessing care in order to elicit their ideas for service improvements. In reality, existing qualitative elicitation and coproduction techniques commonly take more than a year to plan, execute, analyse and report.[20] They require formal ethical review, formally trained qualitative researchers, the use of specialist software and qualitative expertise to interpret and apply the findings.[20 21] These resource requirements are prohibitive for most health system managers, and in many low-income settings there is not a ready supply of specialist expertise.[22] This can lead to well-conducted but one-off engagement activities where the findings are inappropriately generalised to other groups or at the other end of the spectrum are tokenistic and/or methodologically flawed efforts to gather and act on service user feedback, . We are interested in exploring whether it is possible to obtain meaningful and robust findings with rapid tools[23]; here defined as approaches that take 14 days or less 'from entering the field to through to delivery of findings'[24] that is, contacting and recruiting participants, eliciting barriers through the collection and analysis of data, and developing a list of potential interventions to improve the service. Such tools would have very wide application across a broad range of settings and would support the development of PHC-oriented systems that are built on community engagement.[25] While 14 days are essentially arbitrary, it reflects an ambitious target for delivering usable intelligence that aligns with the timescales offered by market research firms to political parties and companies.[26]

## Aim and objectives

We will perform a scoping review[27] of the literature to identify, categorise and evaluate the methods that are being used to rapidly elicit barriers and potential solutions from service users in any community-based health service. We want to understand the strengths and weaknesses of the different methods that have been used, their resource requirements, and their governance frameworks as described by their users.

Responding to the need for rapid, affordable and scientifically robust approaches that can be used to continually improve health services, we ultimately aim to identify the minimum viable product in this space. We want to identify approaches that provide sound, non-tokenistic and actionable intelligence with minimal time, money, equipment, personnel, and skill requirements.

## Review question

What rapid methods have been used to engage with community-based health service users to elicit barriers to access and potential solutions? For each method, what are the main outputs, methodological strengths and limitations and resource requirements in terms of time, personnel and other costs? For the purposes of this review, 'community-based care' will be defined as non-hospital care that involves interaction with a clinician, and a 'community' will be defined as a group who share geographies, interests or identities. This definition is based on that used in the WHO Operational Framework for PHC a.[28] We will use the WHO Operational Framework definition for 'community engagement' that is presented in the introduction.[28] We note that Primary Health Care is not the same as community-based care: the former is a whole-of-society approach to health (that includes hospitals even though it focuses on primary care).[25]

## METHODS AND ANALYSIS
### Guidelines

Our review will be conducted in accordance with the JBI methodology, based on the principles of Arksey and O'Malley and Levac and colleagues.[29–31] Our review will be reported in line with the Preferred Reporting Items for Systematic Reviews and Meta-Analyses (PRISMA) checklist Extension for Scoping Reviews (PRISMA-ScR, online supplemental file).[32] Scoping reviews are the most appropriate method for mapping and characterising the available evidence in a given area, and follow five steps[33–35]:

1. Defining the research question/s
2. Identifying relevant studies
3. Study selection
4. Charting the data
5. Collating, summarising and reporting the results

An iterative approach will be taken towards searching the literature, refining the search strategy, reviewing articles for inclusion, and extracting relevant data.[32 36–38]

## Participants

As we are concerned with barriers to access, we will focus on methods that seek to engage with those who are eligible for a given service but have not managed to attend. As such, we

deem the sample population 'intended service beneficiaries' rather than 'service users'. We will include methods where engagement activities target service users and intended beneficiaries of any community-based health service in any country, serving any need. We will exclude methods that sample exclusively from attendees as—by definition—they have successfully overcome barriers to access.

We will include methods where engagement activities target lay representatives of intended service users such as patient advocacy groups, parents or village elders, however these findings will be reported separately in the findings. We will exclude methods that exclusively engage service providers, managers or policymakers. We will include approaches that engage a mix of users and providers as long as it is possible to disaggregate the findings pertaining to service user engagement.

As we are focusing on groups that face barriers to access, we will exclude approaches that exclusively engage with people who are present at their services, that is, our focus is on methods for contacting and engaging with non-attenders or their proxies.

### Concept

We are interested in methods used for engaging service users to elicit their perceptions of barriers to accessing care and generating ideas for service modifications that could improve access rates and outcomes among people with similar characteristics.

We are focusing on rapid methods, defined as those that can be used to deliver a list of barriers and potential solutions within 14 days or less. This is an arbitrary threshold but draws from our clinical experience in leading health services and represents what we feel to be an acceptable amount of time to generate data to inform real-time decision-making.

Given that it is not standard practice to report the length of time taken to conduct research we anticipate that our search will not identify many studies. To overcome this issue we will include studies that do not state how long they took as long as they meet all other inclusion criteria. We will analyse these studies separately.

We will include all forms of established or novel methods from any scientific field of enquiry. We expect to find examples of the following types of method:

► Interviews: face-to-face, telephone, video call.
► Focus groups
► Group system dynamic modelling
► Q methodology
► Nominal group technique.
► Surveys: in-person, web, telephone, text message
► Rapid ethnography

### Context

We are not limiting the review to any specific population, culture or geography. We will include studies from all countries and any setting except hospital inpatients. Our focus is on in-person access to existing services so we will exclude evaluations of novel services or new interventions.

We primarily define 'access' in terms of whether people are able to physically reach (ie, attend) a clinical provider to get the care they need. This includes attending prebooked appointments as well as presenting to services that do not require appointments. We will include outreach services and home-based care, but exclude virtual/digital remote consults. We will also exclude compulsory care such as when patients are sectioned for mental healthcare, and services where no interaction with a clinician is required, such as automated services to obtain self-testing kits.

### Types of sources

We will include all empirical study types that report on the use of a given method to elicit barriers or potential solutions with a maximum of 14 days between commencing fieldwork and generating the findings.

We will exclude methodological texts, reviews, letters and conference abstracts. We will also exclude systematic reviews, but we will search their reference lists and include any underlying primary studies that meet our inclusion criteria.

### Patient and public involvement

No patient and public involvement.

### Search strategy

The search strategy will be built around rapid community-based methods and access to health services[39 40] (box 1). The search will be limited to human studies published since 1978; the year of the Alma-Ata Declaration on Primary Health Care. The search will be conducted in English but we will include full-text studies published in any language. We plan to complete the review by mid-2023. The search strategy results will be presented in a PRISMA flowchart that will show how studies were eliminated until final search yield that will constitute the basis for synthesis.

We will search the following information resources: the Cochrane Library, MEDLINE Ovid, Embase Ovid and Global Health Ovid. The first 20 pages of Google Scholar will also be screened. The search strategy, including all identified keywords and index terms, will be adapted for each included database and/or information source. Box 1 presents the search strategy for Medline. The Supplementary file (online supplemental appendix) presents the tailored search strategies for all databases. We will check the reference lists of included studies and relevant systematic reviews to identify any additional potentially relevant reports of studies. Key authors will be contacted to uncover additional or upcoming studies.

### Study/Source of evidence selection

Following the search, all identified citations will be collated and uploaded into Covidence (Veritas Health Innovation, Melbourne) and duplicates will be removed. Following a pilot test, titles and abstracts will then be screened by two independent reviewers (HA and RJ) for assessment against the inclusion criteria. Studies that clearly do

**Box 1  Search terms used for Medline**

1. Health Services Accessibility/
2. Health Equity/
3. Social Determinants of Health/
4. (social adj2 determinant adj2 health$).tw.
5. ((health$ or social$ or racial$ or ethnic$) adj5 (inequalit$ or inequit$ or disparit$ or equit$ or disadvantage$ or depriv$)).tw.
6. (disadvant$ or marginali$ or underserved or under served or impoverish$ or minorit$ or racial$ or ethnic$).tw.
7. barrier$.tw.
8. (solution$ or improve$ or strateg$ or access$ or challeng$).ti.
9. Community-Based Participatory Research/
10. Community-Institutional Relations/
11. (communit$ adj3 (engag$ or participat$)).tw.
12. CBPR.tw.
13. (participat$ adj2 health adj2 research).tw.
14. (communit$ adj2 academic adj2 partnership$).tw.
15. (collective adj2 empower$).tw.
16. (equity adj2 mobili$ adj2 partnership$ adj2 communit$).tw.
17. (ethnograph$ or communitarian$).tw.
18. Interviews as Topic/
19. Patient Health Questionnaire/
20. Self Report/
21. Q-Sort/
22. Q-Sort.tw.
23. Q-methodolog$.tw.
24. (system adj2 dynamic adj2 model$).tw.
25. (nominal adj2 group$ adj2 technique$).tw.
26. or/1–25
27. Problem Solving/
28. ((rapid$ or agile) adj2 (appraisal$ or assessment$ or approach$ or evaluation$ or evaluate$ or technique$ or tool$ or method$ or research$)).tw.
29. or/27–28
30. 26 and 29
31. in vitro.tw.
32. (assay$ or microb$).tw.
33. Critical Care/
34. or/31–33
35. 30 not 34
36. limit 35 to humans
37. limit 36 to (comment or editorial or letter)
38. 36 not 37
39. limit 38 to yr='1978-Current'

not meet the inclusion criteria will be excluded. The reviewers will meet after every 10% batch of papers has been screened to discuss any issues. Any disagreements will be resolved through consensus-based discussion, or if necessary, discussion with a third reviewer (LNA).

We will obtain full texts for the potentially relevant papers. The same two review authors will independently assess the papers against the inclusion criteria to determine their eligibility for inclusion. Non-English language papers will be translated into English. The review authors will resolve disagreements through consensus-based discussion, or if necessary, discussion with the same third reviewer. The reviewers will record reasons for exclusion at the full-text screening stage.

The results of the search and the study inclusion process will be reported in full in the final scoping review and presented in a PRISMA flow diagram.[41]

## Data extraction

Two review authors (HA and RJ) will independently extract study characteristics and data from the included studies using a data extraction form developed by the reviewers. The data extraction form will be piloted on three studies by the same two review authors and required amendments will be made by consensus.[42] We anticipate a broad scope of included studies, so data charting will be an iterative process throughout the review. The data extraction tool will be modified and revised as necessary during the process of extracting data from each included evidence source. Any discrepancies will be resolved by group discussion. Modifications will be detailed in the scoping review. Where required, authors of papers will be contacted to request missing or additional data.

The data extracted will include specific details about the participants, concept, context, study methods and key findings relevant to the review question:

► Article title.
► Journal title.
► Authors.
► Country.
► Language.
► Publication year.
► Study type.
► Type of approach (eg, focus group) and description:
  – Setting.
  – Participants.
  – Facilitators.
► Main output if anything other than a prioritised list of potential service modifications.
► Methodological strengths and limitations, as documented by the authors.
► Resource requirements:
  – Number of personnel, and essential skills/level of training.
  – Number of days for each person, full time equivalent.
  – Total number of days taken from conception to findings; including planning, recruitment, engagement and analysis stages.
  – Equipment.
  – Total financial cost.
► Framework used to structure interaction and elicit barriers and solutions.
► Method of recording (notes, audio, etc).
► Other practical requirements or qualitative considerations reported in-text.
► Ethics and governance requirements.
► Level, form, frequency and intensity of participation:
  – Level of participation will be assessed using the five categories used by WHO: inform, consult, involve, collaborate and empower.

- Form will be assessed using the four categories used by WHO: community-oriented, community-based, community-managed and community-owned.
- Frequency is defined as the number of discrete interactions between the project team and the service users.
- Intensity represents the extent to which participants interact, exchange information and influence decision-making in participation processes.[43]

► Power relations, prevailing knowledge and beliefs and cultural barriers,[18] described by the authors.

► Any documented power relations, prevailing knowledge and beliefs and cultural barriers.

## Data analysis and presentation

We plan to conduct a formal narrative descriptive synthesis without meta-analysis. We will stratify the synthesis by methodological approach. We will present a summary table of the different methods used, grouped by discipline. We will also tabulate the resource requirements, form of participation and methodological strengths and limitations. Quantitative resource requirement data will be presented in whole numbers, days and 2022 US dollar amounts as appropriate. Ratios will be used to compare costs between approaches. Qualitative outcomes will be presented narratively. Methods used to engage with service users and service user representatives will be presented separately.

We will not conduct methodological quality assessment of included studies, in keeping with usual practice for scoping reviews.[27 29]

## Limitations

Our review focuses on methods that operate extremely rapidly, using a 14-day cut-off. This choice has driven by our collective experience working with health service and system managers. We are aware that effective community engagement can often take (much) longer than 14 days, and that expediency may come at the cost of the value and nuance of the findings that are delivered. Nevertheless, just because it is unlikely that there are many robust approaches that can deliver meaningful and non-tokenistic findings within a very short timeframe, we feel it is still worth examining the literature to understand this space. There is a risk that rapid approaches produce oversimplified findings that further compound issues for marginalised groups. We will be careful to assess these risks.

**Contributors** LNA conceptualised and planned the study with SK, IG, JE, NT and AB. IG and LNA designed the search terms with input from RJ, HA, SK, JE, JR and NT. LNA wrote the first draft with JR. HA, IG, SK, JE, NT, JR and AB critically revised iterations of the manuscript. All authors read and approved the final protocol.

**Funding** This work was supported by the National Institute for Health Research (NIHR) (using the UK's Official Development Assistance (ODA) Funding) and Wellcome (grant reference 215633/Z/19/Z) under the NIHR-Wellcome Partnership for Global Health Research. The views expressed are those of the authors and not necessarily those of Wellcome, the NIHR or the Department of Health and Social Care. The study was sponsored by the London School of Hygiene & Tropical Medicine. The funders and study sponsor had no role in developing the protocol.

**Competing interests** None declared.

**Patient and public involvement** Patients and/or the public were not involved in the design, or conduct, or reporting or dissemination plans of this research.

**Patient consent for publication** Not applicable.

**Provenance and peer review** Not commissioned; externally peer reviewed.

**ORCID iDs**
Luke Nelson Allen http://orcid.org/0000-0003-2750-3575
Jacqueline Ramke http://orcid.org/0000-0002-5764-1306

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
