## [Reviewer comments · BMJ Open]

ARTICLE DETAILS

TITLE (PROVISIONAL)	Rapid methods for identifying barriers and solutions to improve access to community health services: a scoping review protocol
AUTHORS	Allen, Luke; Azab, Hagar; Jonga, Ronald; Gordon, Iris; Karanja, Sarah; Evans, Jennifer; Thaker, Nam; Ramke, Jacqueline; Bastawrous, Andrew

VERSION 1 – REVIEW

REVIEWER	Edelman, Alexandra James Cook University, Division of Tropical Health and Medicine
REVIEW RETURNED	17-Aug-2022

GENERAL COMMENTS	This aim of the planned review is to map the methods that have been used to rapidly elicit barriers to accessing community health services and identify potential solutions. The proposed review is timely and is likely to yield results that will be useful for policymakers, planners and service administrators interested in exploring methods/approaches to improve service access. The following suggestions to improve the protocol are offered for the authors' consideration: - "Attendance" should be clearly defined in the protocol. It seems as though the implied definition is currently that health care "access" refers only to people attending appointments in clinical settings. It is indeed noted on Page 7 that: "We will exclude services that are delivered remotely". However, health care is increasingly being delivered in settings outside of physical clinics (e.g., homes, virtual consults, outreach services). The protocol should make this definition, and reasons for exclusion of the 'less traditional' service models in the definition of "services" for the purpose of defining "access", clearer early on.- A clearer definition of "Community-based health service" is also needed. In-patient care is often defined as hospital care, yet the term "non-hospital inpatient care" is used. Do the authors instead mean specialist outpatient, primary care, and/or ancillary (e.g., allied health) health care services? Are referral services included in the definition? Or, more narrowly, only clinical services that are owned/governed by community-based entities (rather than government-run or private)?- The authors could add a section in the protocol outlining some likely limitations of the review design. Effective community engagement, for example, can often take longer than 14 days, and it is possible that an emphasis on expediency could come at a cost of research approaches that aim to develop a nuanced understanding of service users' experiences, trust (or lack thereof) in the services offered, or structural health systems issues that contribute to non-attendance. This risk (and potentially other related risks) should be considered carefully given the focus of the
--

	planned review on the most vulnerable and disengaged members of community.
--	--

REVIEWER	Möske, Mike University Medical Center Hamburg-Eppendorf, Medical Psychology
REVIEW RETURNED	01-Dec-2022

GENERAL COMMENTS	Rapid methods for identifying barriers and solutions to improve access to community health services: a scoping review protocol Page / Row Note 4 / 10-14 The authors inform, that qualitative methods are the best way to identify barriers and find solutions. I cannot find any evidence / source for this statement. 6 / 32-36 „For-profit enterprises seem to understand the value of engaging with their customers: many companies use focus groups and market research to continually hone their products and services to meet the evolving needs of their customer base.” □ Reference? 6 / 36 “Our sense is that health programmes are less active in this space.” □ In the introduction section just evidences should be named. 6 / 36-50 References! 6 / 52-57 The concept and the goal of "rapid tools" is not explained enough 8 / 4 It is not absolutely clear what exactly is defined as a participant; are people who are ill but never went to a general practitioner included? Formally they are no service users or non-attenders? Please specify According to the European Commission (Huber et al. 2008) migrant patient groups, older people and people with mental disorders count among those with the greatest barriers in the health care system. Also, other patient groups have been identified as marginalised. Why do you not focus more specific on these groups?
--

	Reference: Huber, Manfred; Stanciole, Anderson; Wahlbeck, Kristian; Tamsma, Nicoline; Torres, Federico; Jelfs, Elisabeth; Bremner, Jeni (2008): Quality in and equality of access to healthcare services. European Commission, Directorate-General for Employment, Social Affairs and Equal Opportunities. 8 / 29 – 32 The justification of the threshold value of 14 days is not well-founded enough or does not seem plausible to me. Why not taken a 2 months period? Also, it is not clear what exactly is submitted in this period? The development of the study material like an interview guideline? The data collection? The data analysis? 8 / 51-52 Why are „hospital inpatients“ excluded? 9 / 20 Specify the timeline 9 / 22 Why are only these databanks used? Why not others like CINAHL Complete 10 / 35 What kind of profession / experiences do the reviewers have? 10 / 42, 53 Are these the same reviewers in the next step? 10 / 43 – 44 “Non-English language papers will be translated into English using Google translate” [ ] very questionable method; is this in line with high quality and accuracy of scientific articles? In this case you have to translate also your search into several languages! How many languages do you plan? Which one? 10 / 57 Switch between active and passive spelling [ ] Why? 11 / 58 – 12 / 5 Here is again active spelling (inconsistent spelling throughout the article)
--	--

REVIEWER	Schultz Petersen , Kirsten
-----------------	----------------------------

	Aalborg Universitet Institut for Medicin og Sundhedsteknologi, Public Health and Epidemiology Group
REVIEW RETURNED	05-Dec-2022

GENERAL COMMENTS	In the abstract you write the following: "We will include studies conducted in any country from 1978 to present. We will not limit by language." I wonder how you will be able to manage that, do you have the available resources to read articles from non-english articles as I do not find the quality of Google translate to be sufficient. On page 8 Participants you aim to include: ".....service users five years and older of outpatient and community-based services in any country, serving any need" this seems very ambitious as it is a very broad target group. I would ask the authors to consider a more narrow target group e.g. adults age 18+. Involving children and youth might be a specific challenge when taking about access and outcome, which I recommend to be carried out in another review. In addition I recommend PPI to be included, as this could help among others in refining the search strategy and discussing the preliminary findings.
---

VERSION 1 – AUTHOR RESPONSE

Reviewer: 1

Dr. Alexandra Edelman, James Cook University

Comments to the Author:

This aim of the planned review is to map the methods that have been used to rapidly elicit barriers to accessing community health services and identify potential solutions. The proposed review is timely and is likely to yield results that will be useful for policymakers, planners and service administrators interested in exploring methods/approaches to improve service access. The following suggestions to improve the protocol are offered for the authors' consideration:

"Attendance" should be clearly defined in the protocol. It seems as though the implied definition is currently that health care "access" refers only to people attending appointments in clinical settings. It is indeed noted on Page 7 that: "We will exclude services that are delivered remotely". However, health care is increasingly being delivered in settings outside of physical clinics (e.g., homes, virtual consults, outreach services). The protocol should make this definition, and reasons for exclusion of the 'less traditional' service models in the definition of "services" for the purpose of defining "access", clearer early on.

- We have added the following text to the introduction: "Complex supply and demand factors govern access to health services and multiple frameworks have been developed, typically defining access as the ability to perceive, seek, reach, pay for and engage with care.^{2,7-11} Access is increasingly being extended through the use of digital services and remote consultations.^{12,13} Whilst these services are useful, they come with their own set of barriers and equity issues, and cannot fully replace the central role played by in-person clinical providers.^{12,14}"

- In the 'Context' section we have clarified that "Our focus is on in-person access to existing services"

- We have also added this section: "We primarily define 'access' in terms of whether people are able to physically reach (i.e. attend) a clinical provider to get the care they need. This includes

attending pre-booked appointments as well as presenting to services that do not require appointments. We will include outreach services and home-based care, but exclude virtual/digital remote consults.”

A clearer definition of “Community-based health service” is also needed. In-patient care is often defined as hospital care, yet the term “non-hospital inpatient care” is used. Do the authors instead mean specialist outpatient, primary care, and/or ancillary (e.g., allied health) health care services? Are referral services included in the definition? Or, more narrowly, only clinical services that are owned/governed by community-based entities (rather than government-run or private)?

- In the ‘Context’ section we state that we aim to include “studies from all countries and any setting except hospital inpatients”. This clearly includes specialist outpatients, primary care, ancillary care, government-owned, community-owned, privately-owned - and any other form of health care that isn’t hospital inpatients. The reviewer’s confusion understandably stems from our clumsy wording in the ‘Review question’ section where we defined community-based care as “non-hospital inpatient care” which can be interpreted two ways. We have changed the text to “non-hospital inpatient care” to clarify. Thanks for picking this up.

The authors could add a section in the protocol outlining some likely limitations of the review design. Effective community engagement, for example, can often take longer than 14 days, and it is possible that an emphasis on expediency could come at a cost of research approaches that aim to develop a nuanced understanding of service users’ experiences, trust (or lack thereof) in the services offered, or structural health systems issues that contribute to non-attendance. This risk (and potentially other related risks) should be considered carefully given the focus of the planned review on the most vulnerable and disengaged members of community.

- We have added the following section:

- “Limitations

Our review focuses on methods that operate extremely rapidly, using a 14 day cut-off. This choice has driven by our collective experience working with health service and system managers. We are aware that effective community engagement can often take (much) longer than 14 days, and that expediency may come at the cost of the value and nuance of the findings that are delivered.

Nevertheless, just because it is unlikely that there are many, if any, robust approaches that can deliver meaningful and non-tokenistic findings within a very short timeframe, we feel it is still worth examining the literature to understand this space. There is a risk that rapid approaches produce oversimplified findings that further compound issues for marginalised groups. We will be careful to assess these risks.”

- We have also added this limitation to the key points.

Reviewer: 2

Dr. Mike Mösko, University Medical Center Hamburg-Eppendorf, Magdeburg-Stendal University of Applied Sciences

Comments to the Author:

Dear authors,

I very much appreciated to read your manuscript on "Rapid methods for identifying barriers and solutions to improve access to community health services: a scoping review protocol". Together with my colleague, Linn Manthey, we both read and reviewed your manuscript. Over and above we very much value your general research question and scope of your study. Your explanations are clearly understandable and transparent. There are just some minor

issues and two more important points - from our point of view - that needed some thoughts and specifications (Participants and language). Enclosed I send you our detailed comments.

I wish you and the team all the best for the review!

Best greetings

Mike Mösko

- Many thanks Mike and Linn

Page / Row	Note
4 / 10-14	The authors inform, that qualitative methods are the best way to identify barriers and find solutions. I cannot find any evidence / source for this statement.  - The journal policy is that abstracts can't be referenced. We have added appropriate references to the introduction where we talk about using qualitative methods.
6 / 32-36	„For-profit enterprises seem to understand the value of engaging with their customers: many companies use focus groups and market research to continually hone their products and services to meet the evolving needs of their customer base.” à Reference?  - Reference added: - Statista. Market research industry [Internet]. 2023 Jan 5. Available from: https://www.statista.com/topics/1293/market-research/
6 / 36	“Our sense is that health programmes are less active in this space.” à In the introduction section just evidences should be named.  - We used this wording because there is no evidence that we are able to cite.
6 / 36-50	References!  - We have added the following references: - Pope C, Mays N. Qualitative Research in Health Care, 4th Edition Wiley [Internet]. Oxford: Wiley Blackwell; 2020 [cited 2023 Jan 5]. Available from: https://www.wiley.com/en-gb/Qualitative+Research+in+Health+Care%2C+4th+Edition-p-9781119410836 - Creswell J, Creswell D. Research Design [Internet]. 5th ed. Thousand Oaks, Calif.: Sage; 2018 [cited 2023 Jan 5]. Available from: https://us.sagepub.com/en-us/nam/research-design/book255675 - Franzen SRP, Chandler C, Lang T. Health research capacity development in low and middle income countries: reality or rhetoric? A systematic meta-narrative review of the qualitative literature. BMJ Open. 2017 Jan 27;7(1):e012332. - Taylor B, Henshall C, Kenyon S, Litchfield I, Greenfield S. Can rapid approaches to qualitative analysis deliver timely, valid findings to clinical leaders? A mixed methods study comparing rapid and thematic analysis. BMJ Open. 2018 Oct 8;8(10):e019993. - Drive Research. How Quickly Can Market Research Be Completed? [Internet]. 2018 [cited 2023 Jan 5]. Available from: https://www.driveresearch.com/market-research-company-blog/how-quickly-can-market-research-be-completed/

6 / 52-57	The concept and the goal of "rapid tools" is not explained enough  - We have updated the text and added a new reference as such: - "rapid tools; here defined as approaches that take 14 days or less "from entering the field to through to delivery of findings"[16] i.e. contacting and recruiting participants, eliciting barriers though the collection and analysis of data, and developing a list of potential interventions to improve the service." - The concept/goals are further unpacked in the aims and objectives section: - "Responding to the need for rapid, affordable, and scientifically robust approaches that can be used to continually improve health services, we ultimately aim to identify the minimum viable product in this space. We want to identify approaches that provide sound, non-tokenistic, and actionable intelligence with minimal time, money, equipment, personnel, and skill requirements."
8 / 4	It is not absolutely clear what exactly is defined as a participant; are people who are ill but never went to a general practitioner included? Formally they are no service users or non-attenders? Please specify According to the European Commission (Huber et al. 2008) migrant patient groups, older people and people with mental disorders count among those with the greatest barriers in the health care system. Also, other patient groups have been identified as marginalised. Why do you not focus more specific on these groups? Reference: Huber, Manfred; Stanciole, Anderson; Wahlbeck, Kristian; Tamsma, Nicoline; Torres, Federico; Jelfs, Elisabeth; Bremner, Jeni (2008): Quality in and equality of access to healthcare services. European Commission, Directorate-General for Employment, Social Affairs and Equal Opportunities.  - Thank you for the reference. We don't want to single out any specific group because the list can never be fully exhaustive. By including all people who are unable to access services (without distinction) we will not run into potential issues caused by erroneous presuppositions. - We are including methods that engage intended service beneficiaries: - We have revised the participant wording to add clarity. - "As we are concerned with barriers to access, we will focus on methods that seek to engage with those who are eligible for a given service but have not managed to attend. As such, we deem the sample population 'intended service beneficiaries' rather than 'service users'. We will include methods where engagement activities target service users and intended beneficiaries of any community-based health service in any country, serving any need. We will exclude methods that sample exclusively from attendees as – by definition – they have successfully overcome barriers to access. We will include methods where engagement activities target service user lay representatives of intended service users such as patient advocacy groups, parents, or village elders, however these findings will be reported separately in the findings. We will exclude methods that exclusively engage service providers, managers, or policymakers. We will include approaches that engage a mix of users and providers as long as it is possible to disaggregate the findings pertaining to service user engagement."

8 / 29 – 32	The justification of the threshold value of 14 days is not well-founded enough or does not seem plausible to me. Why not taken a 2 months period? Also, it is not clear what exactly is submitted in this period? The development of the study material like an interview guideline? The data collection? The data analysis?  - This is a good point, and 14 days is essentially arbitrary – as is 2 months or any other time period we may choose. Our decision to use two weeks comes from years of personal experience working in, with, and running health projects, programmes, and clinics around the world, working in health policy at WHO and the World Bank, and talking with lots of managers who want low cost approaches that can deliver meaningful results in around two weeks or less. There is no external publication we can cite here, however we have tried to clarify our thinking. We would note that political parties and market research firms can conduct large snap surveys and rapid focus groups in a matter of days and want to explore why health has to be an exception? - In terms of what is submitted in this period, the introduction states that we are looking for approaches that take 14 days “to contact and recruit participants, elicit barriers, and develop a list of potential interventions to improve the service“ i.e. the entire research project, from start to finish. - We have added a new limitations section: - “Our review focuses on methods that operate extremely rapidly, using a 14 day cut-off. This choice has driven by our collective experience working with health service and system managers. We are aware that effective community engagement can often take (much) longer than 14 days, and that expediency may come at the cost of the value and nuance of the findings that are delivered. Nevertheless, just because it is unlikely that there are many, if any, robust approaches that can deliver meaningful and non-tokenistic findings within a very short timeframe, we feel it is still worth examining the literature to understand this space. There is a risk that rapid approaches produce oversimplified findings that further compound issues for marginalised groups. We will be careful to assess these risks.” - We have also made this decision:“Given that it is not standard practice to report the length of time taken to conduct research we anticipate that our search will not identify many studies. To overcome this issue we will include studies that do not state how long they took as long as they meet all other inclusion criteria. We will analyse these studies separately”
8 / 51-52	Why are „hospital inpatients“ excluded?  - This is another personal choice – we run community based programmes (primary care and eye screenng) and so are not interested in inpatient services. We are performing the review to inform our own data collection activities.
9 / 20	Specify the timeline  - We have added this information, specifying that we plan to complete the review by mid-2023. As the review period for this protocol is unknown we are unable to provide further detail.
9 / 22	Why are only these databanks used? Why not others like CINAHL Complete  - MEDLINE, Embase and Global Health have been searched for this review. Both MEDLINE and Embase databases cover a broad topic area and have a large

	number of indexed journals. Global Health focuses on international public health. Searching this resource enables us to identify potentially relevant reports that are not indexed on MEDLINE or Embase.  - CINAHL Complete has not been searched for this review as there is a large overlap between its content and that of MEDLINE. When searches are run on CINAHL Complete and the MEDLINE records are removed, there are negligible results remaining, so it was not deemed a useful resource to search.
10 / 35	What kind of profession / experiences do the reviewers have?  - It is not standard practice to include this information in a protocol. For the reviewer's benefit, Hagar Azab is a PHC consultant at WHO with experience working on access to health systems across the Middle East. Ronald Jonga is a registered nurse and public health researcher, currently working as head of audit and clinical effectiveness at a large UK NHS Trust. LA is a family physician, World Bank and WHO health systems consultant, and LSHTM research fellow running trials to improve equitable access to community-based services in LMICs. Between them they have over 70 publications including >10 systematic and scoping reviews published in journals including the Lancet Global Health, BMJ Global Health, and the WHO Bulletin. We don't intent to add this to the text.
10 / 42, 53	Are these the same reviewers in the next step?  - Yes, we have added initials to make this clearer in the manuscript.
10 / 43 – 44	"Non-English language papers will be translated into English using Google translate" à very questionable method; is this in line with high quality and accuracy of scientific articles? In this case you have to translate also your search into several languages! How many languages do you plan? Which one?  - The search will be conducted in English only. We have clarified this in the text. We will include papers that are not written in English. We have updated the protocol to drop the use of Google translate. We will use human translators for this work.
10 / 57	Switch between active and passive spelling à Why?  - This is a writing style preference. We defer fully to the copyeditors to revise the language and grammar.
11 / 58 – 12 / 5	Here is again active spelling (inconsistent spelling throughout the article)  - As above.

Reviewer: 3

Dr. Kirsten Schultz Petersen , Aalborg Universitet Institut for Medicin og Sundhedsteknologi

Comments to the Author:

In the abstract you write the following: "We will include studies conducted in any country from 1978 to present. We will not limit by language." I wonder how you will be able to manage that, do you have the available resources to read articles from non-english articles as I do not find the quality of Google translate to be sufficient.

- We have updated the protocol to remove the use of Google translate. We are fortunate to have won a grant that covers this work and associated studies. We will hire translators if required.

On page 8 Participants you aim to include: ".....service users five years and older of outpatient and community-based services in any country, serving any need" this seems very ambitious as it is a very broad target group. I would ask the authors to consider a more narrow target group e.g. adults age 18+. Involving children and youth might be a specific challenge when taking about access and outcome, which I recommend to be carried out in another review.

- We have used the broadest possible participant eligibility criteria because we do not anticipate that many research studies will have been conducted that meet our other very strict inclusion criteria i.e. methods that take <14 days to "contact and recruit participants, elicit barriers, and develop a list of potential interventions to improve the service."

In addition I recommend PPI to be included, as this could help among others in refining the search strategy and discussing the preliminary findings.

- This particular scoping review is designed to answer a question posed informally but consistently by health service managers. Our co-authorship includes three service managers (Sarah, Nam and Andrew). The final review will directly inform the development of a rapid data collection approach that will be grounded in PPI with marginalised groups.

VERSION 2 – REVIEW

REVIEWER	Edelman, Alexandra James Cook University, Division of Tropical Health and Medicine
REVIEW RETURNED	08-Feb-2023

GENERAL COMMENTS	Thank you for the opportunity to see this manuscript again and to review the authors' responses. The authors have done an excellent job responding to comments and I have no further suggestions.
---

REVIEWER	Schultz Petersen , Kirsten Aalborg Universitet Institut for Medicin og Sundhedsteknologi, Public Health and Epidemiology Group
REVIEW RETURNED	17-Feb-2023

GENERAL COMMENTS	This paper has improved and I look forward to read the review.
--